# Exposure to Stress Alters Cardiac Gene Expression and Exacerbates Myocardial Ischemic Injury in the Female Murine Heart

**DOI:** 10.3390/ijms241310994

**Published:** 2023-07-01

**Authors:** Hemangini A. Dhaibar, Lilly Kamberov, Natalie G. Carroll, Shripa Amatya, Dario Cosic, Oscar Gomez-Torres, Shantel Vital, Farzane Sivandzade, Aditya Bhalerao, Salvatore Mancuso, Xinggui Shen, Hyung Nam, A. Wayne Orr, Tanja Dudenbostel, Steven R. Bailey, Christopher G. Kevil, Luca Cucullo, Diana Cruz-Topete

**Affiliations:** 1Department of Molecular and Cellular Physiology, Louisiana State University Health Sciences Center at Shreveport, Shreveport, LA 71103, USA; hdhaibar@gmail.com (H.A.D.); ldk001@lsuhs.edu (L.K.); nburford3989@gmail.com (N.G.C.); sam001@lsuhs.edu (S.A.); dariocosic05@gmail.com (D.C.); oscar.gomez@uclm.es (O.G.-T.); shantel.vital@lsuhs.edu (S.V.); 2Center for Cardiovascular Diseases and Sciences and Center for Redox Biology and Cardiovascular Disease, LSU Health Sciences Center, Shreveport, LA 71103, USA; xinggui.shen@lsuhs.edu (X.S.); hyung.nam@lsuhs.edu (H.N.); wayne.orr@lsuhs.edu (A.W.O.); tanja.dudenbostel@lsuhs.edu (T.D.); steven.bailey@lsuhs.edu (S.R.B.); chris.kevil@lsuhs.edu (C.G.K.); 3Facultad de Ciencias Ambientales y Bioquímica, Universidad de Castilla-La Mancha, Toledo 45004, Spain; 4Department of Biological Sciences, Oakland University, Rochester, MI 48309, USA; fsivandzade@oakland.edu (F.S.); abhalerao@oakland.edu (A.B.); mancuso@oakland.edu (S.M.); 5Department of Foundation Medical Studies, Oakland University William Beaumont School of Medicine, Rochester, MI 48309, USA; 6Department of Pathology and Translational Pathobiology, Louisiana State University Health Sciences Center at Shreveport, Shreveport, LA 71103, USA; 7Pharmacology, Toxicology and Neuroscience, Louisiana State University Health Sciences Center at Shreveport, Shreveport, LA 71103, USA; 8LSU Health Sciences Center, Department of Internal Medicine, Louisiana State University Health Sciences Center at Shreveport, Shreveport, LA 71103, USA

**Keywords:** stress, glucocorticoids, sexual dimorphism, myocardial infarction, oxidative stress, ferroptosis

## Abstract

Mental stress is a risk factor for myocardial infarction in women. The central hypothesis of this study is that restraint stress induces sex-specific changes in gene expression in the heart, which leads to an intensified response to ischemia/reperfusion injury due to the development of a pro-oxidative environment in female hearts. We challenged male and female C57BL/6 mice in a restraint stress model to mimic the effects of mental stress. Exposure to restraint stress led to sex differences in the expression of genes involved in cardiac hypertrophy, inflammation, and iron-dependent cell death (ferroptosis). Among those genes, we identified tumor protein p53 and cyclin-dependent kinase inhibitor 1A (p21), which have established controversial roles in ferroptosis. The exacerbated response to I/R injury in restraint-stressed females correlated with downregulation of p53 and nuclear factor erythroid 2–related factor 2 (Nrf2, a master regulator of the antioxidant response system-ARE). S-female hearts also showed increased superoxide levels, lipid peroxidation, and prostaglandin-endoperoxide synthase 2 (Ptgs2) expression (a hallmark of ferroptosis) compared with those of their male counterparts. Our study is the first to test the sex-specific impact of restraint stress on the heart in the setting of I/R and its outcome.

## 1. Introduction

In 2019, it was estimated that approximately 20% of adults in the United States suffered from stress-related mental disorders [1]. This number has doubled due to the incertitude experienced during the COVID-19 pandemic [2]. The risk of anxiety, depression, and post-traumatic stress disorder is higher in women; therefore, it is not surprising that the mental health of women was more profoundly affected than that of men^3^ during the pandemic due to a more demanding workload at home and in the workplace and exposure to domestic violence [3]. Severe stress activates neuronal, hormonal, and immune networks, leading to psychiatric symptoms and systemic impacts, including inflammatory, metabolic, and cardiovascular disease effects [4,5,6]. Clinical data suggest that exposure to chronic mental stress doubles the risk for acute myocardial infarction in young women (aged < 50 yrs.) compared with that of men of the same age and similar clinical histories [7,8,9]. It is unknown why mental stress makes women more vulnerable to MI than men.

Restraint stress is a method/model used in animal research that mimics the etiology of psychiatric disorders [10]. Psychiatric disorders are associated with an increased risk of ischemic heart disease in women [11]. However, the molecular and physiological mechanisms underlying the association between psychiatric disorders and AMI are unclear. In the present study, we subjected young adult male and female C57Bl/6J mice (five months of age, which is equivalent to ~30 human years) [12] to a model of restraint stress (mental stress) [10,13] that consisted of restraining the animals for periods of 60 min six times per day for one week (a restraint period equivalent to ~6 human months). Following the exposure to restraint stress, mice were evaluated to assess for behavioral changes and alterations of cardiac gene expression and function. Our findings revealed that female mice subjected to restraint stress displayed significantly increased pathophysiological and behavioral alterations, larger infarct areas in response to ischemia/reperfusion injury, and changes in the gene expression of ferroptosis-related markers, in comparison to their male counterparts.

In conclusion, our study’s results shed light on the sex-specific effects of mental stress on the heart, specifically highlighting the activation of ferroptosis signaling. Moreover, our findings demonstrate that females are more susceptible than males to the detrimental impact of chronic exposure to stress hormones such as corticosterone (the rodent equivalent of cortisol in humans), indicating a heightened cardiac vulnerability in females.

## 2. Results

### 2.1. Restraint Stress Significantly Increases Corticosterone Levels with More Pronounced Changes in Stressed Females

Restraint stress caused a significant increase in corticosterone levels in restraint females when compared to both control females and restraint male subjects (Figure 1A) (656.55 ± 98.74 ng/mL vs. 275.6 ± 54.13 ng/mL and 332.13 ± 27.86 ng/mL, respectively). While restraint stress did not result in a statistically significant increase in corticosterone levels in males, we observed an increase in the average value, indicating that restraint stress in our model leads to biologically significant differences in corticosterone production in both males and females to a similar extent (Figure 1A).

### 2.2. Restraint Stress Leads to Behavioral Changes That Mimic Stress Disorders in Humans

The plus maze test was used to assess the behavior changes in all the experimental animals 24 h after the 7-day exposure to restraint stress. Results from the restraint groups were cross compared against the corresponding controls. Specifically, this test assesses anxiety-related (aversion to elevated open areas) and instinctive exploratory behaviors in novel environments [14] in rodent models. Our data showed significant behavioral differences between control and restraint female mice following one week of chronic exposure to restraint-induced stress (Figure 1B). Whereas female control mice stayed in the open arms of the maze for 80.67 ± 14.91 s, that time was reduced to just 22.4 ± 14.91 s for restraint females (Figure 1B). By contrast, the time that male control mice spent in the open arms of the maze was not significantly different from that measured for their restraint counterpart (Figure 1B).

Restraint female mice also displayed impairments in the startle response (Figure 1C). Restraint female mice showed a significantly decreased startle amplitude (107.6 ± 76.15 s) compared with restraint male mice (204.12 ± 35.33 s).

### 2.3. Restraint Stress Affects the Cardiac Expression of Hypertrophy and Inflammatory Markers

We wanted to determine whether exposure to restraint stress promotes cardiac-specific genomic changes despite the absence of detectable cardiac dysfunction by measuring the expression levels of genes associated with pathological cardiac hypertrophy and heart failure (Figure 2). Our data show that restraint stress upregulated the transcription of fetal genes associated with pathological cardiac hypertrophy—skeletal muscle α-actin (Ska), β-myosin heavy chain (β-Mhc), and brain natriuretic peptide (Nppb)- in R-male hearts compared to their control counterparts (Figure 2A). However, only Nppb showed significant up-regulation in R-female hearts compared to control female hearts (Figure 2A). In contrast, the upregulation of inflammatory markers, including interleukin-6 (Il-6) and lipocalin 2 (Lcn-2), was significantly pronounced in S-female hearts but not in their male counterparts (Figure 2B).

### 2.4. Restraint Stress Increases Sexual Dimorphism in Cardiac Gene Expression and Leads to the Expression of Genes Involved in Cell Death and Reactive Oxygen Species Production

We performed a genome-wide microarray analysis of hearts harvested from controls and restraint male and female mice to characterize further the changes in cardiac gene expression triggered by stress. Our data showed that although at baseline (NS), 77 genes differed in expression between male and female hearts, exposure to stress increased the number of sexually dimorphic genes to 119 (Figure 3A). Only 25 genes were sexually dimorphic independently of the treatment. In control hearts, 52 unique genes were differentially expressed between males and females, and the expressions of 94 genes were sexually dimorphic only in restraint hearts (Figure 3A). There were no significant differences in gene classification of the differentially expressed genes between control and restraint conditions. Most of the identified sexually dimorphic genes were enzymes and G-protein coupled receptors (Figure 3B).

Analysis of the differentially expressed genes between restraint and control hearts suggests that, at large, restraint stress acts as a promoter for gene expression (shown in orange) (Figure 4A). Further analysis of pathways, diseases, and physiological processes indicated that exposure to stress triggers sexual dimorphism in the expression of genes involved in healing, cell migration, cell growth, ROS synthesis, and cell death (ferroptosis) (Figure 4B). Focusing in particular on ferroptosis, we identified several genes associated with this form of programmed cell death, including tumor protein p53 (p53), solute carrier family 38 member 1 (Slc38A1), dexamethasone-induced Ras-related 1 (RasD1), cyclin-dependent kinase inhibitor 1A (Cdkn1A or p21) and angiopoietin-like 4 (Angptl4) (Figure 4B).

### 2.5. Restraint Stress Exacerbates Myocardial Infarction Injury in the Female Heart but Not in the Male Heart

To test whether the changes in gene expression triggered by restraint stress translate into a worse outcome following myocardial infarction and to test if the effects were more profound in females than in males, we subjected controls and restraint mice (both genders) to 60 min of ischemia followed by reperfusion for 48 h simulating a myocardial infarction (Figure 5A). A parallel cohort of gender-matched control and restraint mice underwent a sham procedure (see Materials and Methods section). Although stress appears to have significant effects on the number of ischemic areas in sham-operated mice in female hearts (Figure 5A), no significant effects on the size of the infarct area were observed in their male counterparts (Figure 5A). Similarly, stress led to a statistically significant increase in the infarct area in restraint-ischemia/reperfusion female hearts compared with their control-ischemia/reperfusion-counterparts (Figure 5A). No significant differences were found in male hearts. Consistent with these results, H&E and Masson’s trichrome staining showed a more prominent infarct area, signs of edema and inflammation, and fibrosis in heart sections from proestrus-stressed female mice challenged with ischemia/reperfusion compared with their controls and the corresponding male counterparts (Figure 5B).

### 2.6. Restraint Stress Modulates the Expression of Mediators of Ferroptosis in the Heart in a Sex-Specific Manner and Promotes Sex-Specific Changes in Mediators of Ferroptosis in the Heart

We evaluated changes in p53 and p21 (Cdkn1A) at the protein level (Figure 6A,B). Our microarray analysis suggested that exposure to stress significantly affects the levels of p53 in the heart. Under baseline conditions, p53 was almost undetectable in control male hearts (Figure 6A).

In contrast, p53 levels were significantly elevated at baseline in control female hearts compared to their male counterparts (Figure 6A). Under stress conditions, p53 levels became detectable in the sham restraint male heart. As suggested by the microarray, the levels were significantly higher than those observed in their restraint female counterparts (Figure 6A). Notably, restraint stress had the opposite effect on female hearts, significantly reducing p53 expression levels (Figure 6A). Ischemia/reperfusion did not further alter the levels of p53 in the restraint groups (Figure 6A).

Consistent with our microarray data, p21 was significantly elevated in sham restraint male hearts compared to their female counterparts (Figure 6B). Under ischemia reperfusion conditions, p21 levels were significantly reduced in restraint male hearts compared to the corresponding sham group (Figure 6B). The opposite was noted in female hearts where under ischemia/reperfusion conditions, p21 levels in restraint females were significantly increased compared to ischemia/reperfusion controls and sham restraint females (Figure 6B).

### 2.7. In the Female Heart, Restraint Stress Decreases the Expression Levels of Nrf2 and Increases Lipid Peroxidation, Superoxide Levels, and the Expression of Ptgs2

Nuclear factor erythroid 2–related factor 2 (Nrf2) is one of the central regulators of the antioxidant response [15] and a target of p53 regulation [16]. Nrf2 levels were elevated in control female hearts at baseline compared to their R counterparts and remained elevated post-I/R (Figure 7A). Restraint stress appeared to decrease the levels of Nrf2 in females under sham operated and I/R conditions, while no effects were observed in male hearts under any treatment (Figure 7A). GPX4 is a downstream effector of Nrf2 and catalyzes the reduction of hydrogen peroxide, hydroperoxides, and lipid hydroperoxides (a hallmark of ferroptosis), thereby protecting cells against oxidative damage. Although no significant changes were found in GPX4 levels, we observed a significant increase in lipid peroxidation in R-female hearts compared to their corresponding controls and male counterparts (Figure 7B). Consistent with the increase in lipid peroxidation, superoxide production was exacerbated by stress in the female heart in response to I/R (Figure 7B). Prostaglandin-endoperoxide synthase 2 (Ptgs2) is a marker for ferroptosis, and an increase in its expression correlates with lipid peroxidation. Our data showed that Ptgs2 expression was significantly upregulated in I/R in both control and R-male hearts (Figure 7C); however, in females, only the stress groups showed significant increases in Ptgs2 expression (Figure 7C).

## 3. Discussion

Exposure to chronic mental stress has been identified as a risk factor for myocardial infarction in women [7,8]. Women have an elevated risk of cardiovascular complications associated with stress [17,18,19,20]. However, the molecular mechanisms underlying the increased cardiovascular risk in women are unknown. In our study, we found that 5-month-old (the equivalent of a human age of ~30 years) [12] female mice exposed to restraint stress showed more persistent behavioral and physiological changes than their male counterparts (Figure 1). Our data also showed that corticosterone levels were significantly increased only in restraint female mice compared to restraint male mice after one week of restraint stress (Figure 1). However, no differences in adrenocorticotropic releasing hormone (ACTH) were found in either group. Studies have shown that an increase in glucocorticoid levels activates glucocorticoid-mediated negative feedback for the termination of the HPA axis response to stress by inhibiting the release of corticotropin-releasing hormone and ACTH from the paraventricular neurons and anterior pituitary gland, respectively [21]. Therefore, in our model, the high corticosterone levels most likely signaled back to the pituitary gland to repress ACTH (negative feedback loop) in an attempt to repress glucocorticoid production by the adrenal gland and control the stress response, which explains why no significant differences in ACTH were detected between controls and R mice. These findings are also consistent with published human studies showing that females are more susceptible and reactive to stress than males of similar age and clinical characteristics [22,23,24,25].

We found no detectable cardiac abnormalities by an echocardiogram after subjecting the animals to the restraint stress challenge. However, we only measured systolic function, and the measurements were performed on anesthetized mice, which may mask functional differences between treatments. It has been documented that anesthetics influence cardiac function [26]; therefore, it is possible that no cardiac abnormalities were detected in our model due to this limitation. Another limitation of this study is that the evaluation of cardiac function was not comprehensive, and it is possible that diastolic function was significantly compromised in response to restraint stress in females. Previous studies have demonstrated that mental stress can adversely affect resting left ventricular (LV) diastolic function in patients with heart failure [27] and in women with posttraumatic stress disorder [28], therefore, it is plausible that similar changes in diastolic function could be observed in our model. Future studies are necessary to thoroughly investigate the effects of restraint stress on cardiac function and provide a more detailed characterization.

Gene expression analysis of hearts harvested from control and restraint mice revealed changes in the expression patterns of genes involved in pathological hypertrophy and inflammation (Figure 2). Cardiac hypertrophy serves as an adaptive response to increased workload. However, when this increased workload becomes chronic, the compensatory hypertrophy transitions into pathological hypertrophy. Chronic stress exposure has been linked to an elevated risk of pathological cardiac hypertrophy, primarily due to sustained cardiac workload increments that induce a chronic reactivation of genes associated with fetal development. This reactivation triggers increased protein synthesis, leading to cardiomyocyte growth and remodeling of the cardiac sarcomeres, resulting in structural and mechanical alterations.

β-myosin heavy chain (myh7), skeletal muscle α-actin (Acta1), and brain natriuretic peptide (Nppb) are predominantly expressed in the heart during embryonic and fetal stages, with their expression significantly reduced in ventricles after birth. However, in pathological cardiac hypertrophy, the expression of these genes is markedly upregulated. Previous studies have indicated that imbalances in glucocorticoid signaling in cardiomyocytes are associated with the expression of these markers. Furthermore, their expression shows sexual dimorphism, with higher levels observed in male hearts compared to female hearts. Consistent with these findings, our restraint model exhibited statistically significant increases in myh7, Acta1, and Nppb expression in restraint male hearts compared to their non-stressed (controls) counterparts. However, in restraint female hearts, only Nppb showed a significant elevation. Restraint female hearts displayed more pronounced and significant changes in the expressions of Il-6, and Lcn-2 (Figure 2). Il-6 plays a central role in heart failure (HF) pathophysiology. A recent clinical study showed that elevated Il-6 levels are found in more than 50% of HF patients with a poor prognosis [29]. Increased expression of Lcn-2 in the heart has been associated with cardiomyocyte apoptosis and the development of pathological cardiac hypertrophy and progression to HF [30]. Published data suggest that glucocorticoid receptors regulate Lcn2 expression in cardiomyocytes, and alterations in its expression are implicated in the progression to pathological cardiac hypertrophy. In our study, we observed significant upregulation of Il-6 and Lcn2 expression in restraint female hearts compared to their controls, following restraint stress. These findings indicate that stress may induce cardiac pathology by triggering sex-specific gene expression patterns in the heart.

Genome-wide gene expression analysis (Figure 4) of the hearts from male and female control and restraint mice also showed that ferroptosis and increased levels of reactive oxygen species were among the top ten predicted dysregulated canonical pathways and biological functions (Figure 5). Among the genes within the ferroptosis category, we found that the cardiac levels of p53, and Cdkn1A (p21).

Changes in the activation status of the p53–p21 pathway have been reported to modulate cell death and the antioxidant response in disease states [31]. Studies have shown that the levels of p53 are susceptible to stress stimuli such as DNA damage, oncogene expression, ribosomal dysfunction, hypoxia, and oxidative stress. p53 has been shown to promote cell death, cell cycle arrest, metabolism alterations, and autophagy [32,33]. Because of these activities, p53 has been studied as a potent antitumor regulator. However, the role of p53 in the heart has gained recent attention [34]. At baseline, p53 is essential for embryonic cardiac development and maintaining normal cardiac architecture, metabolism, and mitochondrial function [35]. In pathological conditions, including cardiac hypertrophy, chemotherapy-induced cardiotoxicity, diabetic cardiomyopathy, and myocardial infarction (MI), studies suggest that p53 negatively impacts oxidative stress and cardiomyocyte apoptosis [36,37]. Some studies have shown that p53 activation promotes cell death by inhibiting Nrf2 and downregulating the expression of the cysteine/glutamate antiporter SLC7A11 (which imports cysteine for glutathione biosynthesis and antioxidant defense) [38]. However, some studies also show that p53 can exert cardioprotective effects by activating antioxidative genes, including p21, which in contrast to p53, has a well-established cardioprotective role (protects cells from oxidative stress injury) [39]. p21 protects the heart from oxidative stress by suppressing endoplasmic reticulum stress [40] and stabilizing Nrf2 [41]. A recent study found that inhibition of p53 suppressed p21 expression in cardiomyocytes, and the decrease in p21 led to the downregulation of Nrf2, which blocks the expression of the antioxidant response gene in cardiomyocytes [42]. Activation of the p53–p21 pathway has also been shown to inhibit ferroptosis by suppressing the accumulation of toxic lipid reactive oxygen species and promoting the conservation and biosynthesis of the cysteine-derived antioxidant glutathione (GSH) via p21 [43,44,45,46]. Therefore, p53 may represent a double-edged sword for the heart, with its positive effects possibly mediated by p21. Our results also suggest that the activity of the p53-p21 pathway may be dependent on the physiological context.

Overall, our microarray data suggest that the expression of these ferroptosis genes in the heart is modulated by stress, and their expression is sexually dimorphic. Additionally, the data suggest that ferroptosis may be a predominant cell death program in the heart activated by mental stress. To further test whether mental stress acts on the expression of these genes and predisposes the heart to a worse outcome after myocardial infarction (MI), we challenged male and female mice in our stress and ischemia reperfusion models.

Exposure to stress only exacerbates the ischemic damage in ischemia/reperfusion challenged females: increased infarct areas and more pronounced morphological changes, including inflammatory cell infiltration, myofiber disarray, and fibrosis (Figure 5). These results are consistent with the effects of stress on the levels of p53 and p21 (Figure 6). The p21 band was detected at a higher molecular weight (~26 kDa) in the figure. This observation suggests that our antibody may be detecting phosphorylated p21, indicating potential changes in its activation status. Further studies should be conducted to determine whether exposure to stress influences the phosphorylation status of p21 in the heart. Prior to our study, no data were available detailing the association between mental stress and the sex-specific dysregulation of these genes and proteins in the heart. We found that p53 expression is sexually dimorphic in the heart, and although its levels are almost undetectable in males, female hearts expressed high p53 levels. The harmful or cardioprotective role of p53 may also be sex dependent. Increases in the p53 response to stress in the male heart are consistent with published studies showing a negative role for p53 in MI. In males, the p53 increase leads to more pronounced ischemic damage under control conditions with ischemia/reperfusion compared to their female counterparts.

In contrast, elevated p53 at baseline in control females protects the heart against ischemic injury. Restraint stress-induced changes in p53 expression could play a crucial role in mediating the augmented cell death observed in female hearts following ischemia/reperfusion. Further investigations are warranted to elucidate the involvement of p53 signaling in the sex-specific cardiac response to stress.

The results on the changes in p53 levels correlate with our finding that Nrf2 is elevated at baseline in the female heart, suggesting a more robust antioxidant response. Nrf2 is an essential transcriptional regulator of anti-ferroptosis genes, preventing lipid peroxidation and free iron accumulation. Decreases in Nrf2 sensitize cells to ferroptosis [47]. We found that Nrf2 levels were significantly decreased by stress only in female hearts (Figure 7). The lower Nrf2 levels in restraint female hearts correlated with increased superoxide levels in ischemia/reperfusion restraint female hearts (Figure 7), suggesting increased oxidative stress. The production of ROS was significantly elevated in ischemia/reperfusion restraint female hearts compared to their controls and ischemia/reperfusion restraint male hearts (Figure 7). Lipid peroxidation, one of the main biochemical hallmarks of ferroptosis [48], was also significantly increased in restraint female hearts compared to their control counterparts. However, we did not find statistically significant increases in lipid peroxidation in ischemia/reperfusion restraint female hearts compared to their ischemia/reperfusion controls. Instead, we observed a significant increase in lipid peroxidation in ischemia/reperfusion restraint female hearts compared to ischemia/reperfusion restraint male hearts (Figure 7). Further studies will be needed to clarify the role of p53, p21, and Nrf2 in the stressed female heart.

Ptgs2 is an inflammatory mediator associated with ferroptosis [49], and the upregulation of Ptgs22 mRNA is used as a pharmacodynamic marker of ferroptosis in tissues [50]. Consistent with the changes in Nrf2, ROS, and lipid peroxidation, we also found that Ptgs2 is significantly increased by stress following ischemia/reperfusion in female hearts. Future studies are needed to define the mechanisms by which stress leads to these molecular and biochemical effects on the heart and to dissect whether these effects are directly mediated by the glucocorticoid receptor signaling in cardiomyocytes and whether changes in p53-p21 levels result from glucocorticoid receptor’s inhibition of estrogen signaling in the heart as our previously published data seem to suggest.

## 4. Materials and Methods

### 4.1. Reagents and Antibodies

The anti-ERα and -β antibodies were purchased from Santa Cruz Biotechnology (Dallas, TX). Anti-GR (D6H2L XP^®^ rabbit mAb #12041) and anti-p53 (1C12 mouse mAb #2524) antibodies were purchased from Cell Signaling (Danvers, MA, USA). Anti-p21 (Waf1/Cip1/CDKN1A p21 Antibody (F-5) sc-6246) was purchased from Santa Cruz Biotechnology (Santa Cruz, CA, USA). Anti-Nrf2 (ab92946) and anti-GPX4 (ab125066) antibodies were purchased from Abcam (Cambridge, UK). Lastly, 5-Ethyl-5,6-dihydro-6-phenyl-3,8-diaminophenanthridine (DHE; cat. no. 37291) and dimethyl sulfoxide (DMSO; cat. no. 472301) were purchased from Sigma–Aldrich (St. Louis, MO, USA).

### 4.2. Animals

Five-month-old male and female C57BL/6 mice (strain #:000664) were purchased from the Jackson Laboratory (Benton Harbor, ME). The mice were shipped to our facility, and upon arrival, all animals were maintained following the LSU Health Sciences Center Shreveport directives for the care and use of laboratory animals. Following one week of acclimation to the new environment, each group of male and female mice was randomized into control and restraint animals. Female mice were housed in groups of four to synchronize their estrous cycle. Vaginal smear cytology was used to determine the estrous cycle phases. Both groups were brought to the laboratory every morning between 8 and 9 am and placed in two different laboratory sections.

Control mice were left in their cages (covered and placed in a quiet laboratory area). Instead, the animals in the restraint group were removed from their cages and subjected to repeated cycles of restraint stress for a week (six cycles/day; 60 min/cycle with 30 min intervals between each cycle). The restraint protocol included placing the animals on a plastic platform restricting their upper body (except the head) and limb movements with doubled-sided Velcro tape (Scotch fasteners). The tail was also restrained using nonadhesive medical tape (Nexcare for sensitive skin). When the restraint animals were subjected to restraint cycles, the unrestraint control animals were prevented from accessing food and water. During the 30 min interval between each cycle, the restraint mice were returned to their cages, and free access to food and water was restored for both restraint and control animals. At the end of the last daily restraint cycle (at approximately 5 pm), all mice were returned to the animal facility for the night. Following a week (five days) of exposure to restraint stress, the mice were evaluated for behavioral changes, and their cardiac function was assessed by echocardiogram.

### 4.3. Corticosterone and Estradiol Measurements in Blood

Blood samples were collected by tail bleeding. Briefly, mice were gently restrained on a heating pad by holding the base of the tail with the forefinger and thumb of one hand. One to two millimeters of the tip of the mouse tail was cut using sharp, sterile scissors. The tail was then firmly stroked from the base to the tip of the tail using the forefinger and thumb of the other hand. Approximately 100 μL of blood was collected into a 200 μL EDTA-coated capillary tube (Hopkins Medical Products, Grand Rapids, MI). Following this procedure, gentle pressure was applied to the tail with sterile gauze. Blood Stopper Powder (DOGSWELL; Remedy + Recovery) was applied immediately to stop the bleeding. After ensuring the blood flow had stopped, the mouse was returned to its cage. Blood samples were processed by centrifugation (3000 rpm for 15 min at 4 °C), and the plasma was separated for further analysis. R mice were bled at day 0 (before starting the restraint challenge) and at the end of the last restraint cycle on day 7. Blood samples were collected from control mice in parallel with the restraint animals.

Plasma corticosterone (catalog #K014-C1) and 17β-estradiol (catalog #KB30-H1) levels were measured using commercially available ELISA kits (Arbor Assays, Ann Arbor, MI). The corticosterone assay sensitivity and detection limits were 6.71 pg/mL and 12.8 pg/mL, respectively. The sensitivity and detection limits for the 17β-estradiol ELISA were 2.21 pg/mL and 2.05 pg/mL, respectively, as reported by the manufacturer. The respective intra- and inter-assay coefficients of variation (CVs) were 9.4–15.1% for the corticosterone assay. For 17β-estradiol, the intra- and inter-assay CVs were 3.9–7.3% and 3.6–13.8%, respectively. All samples were assayed in duplicate following the manufacturer’s instructions.

### 4.4. Left Anterior Descending Artery Ligation Procedure

Ischemia/reperfusion injury was as follows: each mouse was placed in an induction chamber and anesthetized using 5% isoflurane and oxygen with a flow rate of 0.4 L/min until loss of the righting reflex and then maintained with 2% isoflurane in 100% oxygen with a flow of 0.4 L/min (ventilation) using a nose cone tube connected to the anesthesia apparatus until the tracheal tube was inserted. Following intubation, the animal was placed on a surgical platform, and the chest was shaved and prepared with betadine and alcohol before the incisions were made. The left anterior descending artery was identified on the surface of the heart using a dissection microscope and then ligated by passing a 6–0 silk suture underneath the LAD and making a loose double knot with the suture, leaving a 2–3 mm diameter loop through which a 2–3 mm long piece of polyethylene (PE-10 tubing was placed. The loop was tightened around the artery and tubing. The occlusion of the left anterior descending artery was confirmed by the appearance of a paler color on the anterior wall of the left ventricle. After 60 min of ischemia, the knot was untied, and the PE-10 tubing was removed. Reperfusion was confirmed by the appearance of a pink-red color after 15–20 s. The chest cavity was then closed by sewing the incision in the third intercostal space with a 4–0 silk suture. When suturing was completed, the isoflurane flow was stopped, the mouse was removed from the ventilator, and the tube was carefully removed. The mouse remained under observation for 5 min and then returned to the cage for recovery. Animals were continuously monitored for distress or pain during the 48-h following reperfusion. The analgesic carprofen (5 mg/kg administered subcutaneously) was used for pain relief from the surgery. The sham left anterior descending artery procedure involved the surgical preparation of the animal without ligation, followed by the same duration of reperfusion (48 h) as the experimental group. After anesthetizing the mouse, a small incision was made in the chest to expose the heart. The heart was then kept exposed for 60 min without ligation. Following this period, the chest was closed, and the mouse could recover and receive post-operative care similar to the mice that underwent the procedure.

Smears were performed, and female mice were classified according to their phase in the estrous cycle as at estrus (low basal estradiol levels) or proestrus (high estradiol levels) before undergoing the ischemia/reperfusion procedure.

These studies were approved by the LSU Health Sciences Center Animal Care and Use Committee (Protocol P23-004).

### 4.5. Elevated Plus Maze Test and Startle Response Measurements

An elevated plus maze test was used to measure anxiety-like behavior in our experimental groups. The plus-shape maze was elevated 60 cm off the floor and had two open arms (37 × 8 cm) and two closed arms (37 × 8 × 14 cm) extending from a common central platform (8 × 8 cm). Each mouse was placed on the central platform and allowed to explore for 5 min. Each arm’s number of entries and dwelling time were analyzed using a video-tracking system (EthoVision XT, Noldus, VA). Following each session, the maze was cleaned with 70% ethanol to remove odors.

All mice were allowed to acclimate for one hour in the experimental room for the startle response measurements on the test day. Following this acclimation period, mice were subjected to prepulse inhibition (an inducible neurological phenomenon in which a weaker sensory prestimulus attenuates/inhibits the motor reflex response or “startle reaction” of the test subject to a subsequent strong reflex-eliciting stimulus). Each animal was individually placed within a Plexiglas cylinder (12.7 cm long × 1.5 cm in diameter) resting on a Plexiglas frame (12.7 cm × 20.3 cm) housed within the startle response chamber (38.1 cm × 40.6 cm × 58.4 cm, San Diego Instruments, San Diego, CA, USA). Startle amplitudes were detected and measured by a piezoelectric accelerometer mounted directly below the animal’s midline on the Plexiglas frame’s underside. Mice were allowed to acclimate to the startle response chamber for 5 min before the onset of a PPI session. The background noise intensity was 65 dB, whereas the startle pulse intensity was 120 dB. PPI sessions consisted of 58 trials. Each trial was followed by an intertrial interval (ITI) with a duration averaging 15 s. The PPI session consisted of eight distinct trial types that were pseudorandomly presented five different times during the session. These include (1) no stimulus; (2) startle pulse alone; (3) prepulse alone with prepulse intensities of 4, 8, or 16 dB (PP4, PP8, and PP16, respectively) above background (65 dB); and prepulse (PP4, PP8, and PP16) with startle pulse trials (120 dB). All prepulse and startle pulses had durations of 20 ms and 40 ms, respectively. For prepulse with startle pulse trials, the interval between the onset of the prepulse and the startle pulse was 100 ms. Startle responses were collected for each trial throughout each mouse subject’s PPI session. The startle amplitude was calculated as the maximum motion index during the 100 ms after the pulse onset minus the maximum motion index during the 100 ms before the onset of the prepulse. Eight mice per group were used in all the experiments.

### 4.6. Heart Histology

Mice were euthanized with an overdose of isoflurane, and whole hearts were sliced using an acrylic mouse heart slicer matrix. The hearts’ coronal sections (1.0 mm approx.) were used for different experimental protocols. Consistently across all the experimental animals, the second and third slices from the heart’s apex were used for 2,3,5-triphenyltetrazolium chloride staining (TTC, Catalog T8877, Sigma, St. Louis, MO, USA) and other histological exams, respectively. The heart slices were incubated in 1% TTC at 37 °C for 10 min, then placed between two glass slides and scanned. A red color indicates live tissue, whereas white/pale zones denote infarct areas which were quantified using ImageJ color threshold mode to differentiate between injured and viable cardiac tissue (https://imagej.nih.gov/ij/docs/menus/analyze.html, accessed on 10 March 2022). The heart slices marked for histological examinations were transferred to histological cassettes (Leica Biosystems #3802765) and fixed in 10% formalin overnight. Formalin-fixed, paraffin-embedded tissue blocks were sectioned at 5 µm, deparaffinized in xylene, hydrated in a graded ethanol series, rinsed with distilled water, and washed with PBS. Sections were stained for hematoxylin-eosin (H&E) and Masson’s trichrome (MT).

### 4.7. RNA Extraction and Quantitative Real Time-PCR (qRT–PCR)

Total RNA was isolated from tissues and cells using the RNeasy Mini Kit and RNase-Free DNase Kit (Qiagen, Valencia, CA, USA) according to the manufacturer’s instructions, with deoxyribonuclease (DNase) treatment performed on the column. RNA purity and yield were assessed by evaluating the A260/A280 ratio and concentration using a NanoDrop One Spectrophotometer (Thermo Fisher Scientific, Waltham, MA, USA). The One-Step RT–PCR Universal Master Mix reagent (Thermo Fisher Scientific) was used to quantify the target gene mRNA levels. Quantitative real-time polymerase chain reaction (qRT–PCR) was performed with the CFX96 Real-Time System C1000 Touch Thermal Cycler (Bio-Rad) using predesigned primer–probe sets (Thermo Fisher Scientific) for β-myosin heavy chain (β-Mhc) (Myh7b, Mm01249941_m1), skeletal muscle α-actin (Ska) (Acta1, Mm00808218_g1), brain natriuretic peptide (Nppb) (Mm01255770_g1), interleukin-6 (Il-6) (Mm00446190_m1), lipocalin 2 (Lcn-2) (Mm01324470_m1), prostaglandin-endoperoxide synthase 2 (Ptgs2) (Mm00478374_m1), and the reference gene peptidylprolyl isomerase B (PPIB). The thermocycling parameters for each reaction were 48 °C for 30 min and 95 °C for 10 min, followed by 40 cycles of 95 °C for 15 s and 60 °C for 60 s. Values measured for each primer–probe set were normalized to PPIB.

### 4.8. Microarray Studies

The microarray data are available in the Gene Expression Omnibus repository at the National Center for Biotechnology Information. Twenty-four RNA samples (12 males and 12 females, treatments nonstressed (controls) and exposed to restraint stress [R]) were assessed and processed for hybridization with Clariom S Mouse arrays. RNA quality was determined with the Agilent TapeStation RNA assay (Agilent Technologies, Santa Clara, CA, USA). RNA quantity was assessed with the Qbit Broad Range RNA assay (Invitrogen, Waltham, MA, USA). RNA samples were processed and labeled for hybridization according to the standard GeneChip WT PLUS Reagent Kit manual for target preparation for GeneChip Whole Transcript (WT) Expression. Approximately 150 ng of fragmented, biotin-labeled, sense-strand ss-cDNA was hybridized to Affymetrix GeneChip Clariom S mouse arrays. For the data analysis, pixel intensity measurement, feature extraction, data summarization, normalization, and differential gene analysis were performed in the Transcriptome Analysis Console (TAC version 4.0, Thermo Fisher Scientific). An ANOVA was used to identify significant differentially expressed genes between the groups (males versus females in no stress and restraint stress conditions). Only genes with a fold-change of −2< or >2 were considered for the analysis. Arrays were normalized using the SST-RMA (Signal Space Transformation Robust Multi-Chip Analysis) algorithm, which consists of background adjustment, quantile normalization, and summarization. The list of probe sets generated were visually sorted using a Venn diagram generator and further analyzed with Pathway Analysis version 6.5 (Ingenuity Systems).

### 4.9. Western Blotting

Tissues were harvested and mechanically lysed by bead homogenization and sonication using RIPA buffer (Sigma R0278) with protease (ProBlock™ Gold; GoldBio Catalog #GB-108) and phosphatase (Simple Stop™ 1; GoldBio Catalog #GB-450) inhibitors. After incubation on ice for 30 min, the supernatant was harvested by centrifugation at 12,000 rpm for 20 min at 4 °C, followed by protein concentration measurements using the BSA method. Approximately 70–90 μg of protein were loaded and separated by 10% sodium dodecyl sulfate-polyacrylamide gel electrophoresis and transferred to PVDF membranes (Bio-Rad, Hercules, CA, USA). These membranes were first blocked in Pierce™ Blocking Buffer (PI37570 Protein Free, Thermo Scientific, Waltham, MA, USA) for 1 h, washed three times with Tris-buffered saline Tween (TBST, pH 7.6), and finally incubated overnight at 4 °C with antibodies against the following proteins: p21 (sc-6246; Santa Cruz, CA, USA; 1:200), p53 (1C12 Mouse mAb; Cell Signaling Technology, Danvers, MA, USA; 1:1000), Nrf2 (ab92946, rabbit polyclonal; Abcam, Cambridge, UK; 1:1000) and GPX4 (ab125066, rabbit monoclonal; Abcam, Cambridge, UK; 1:1000). After washing, the membranes were incubated with species-specific secondary HRP-conjugated antibodies for 1 h at room temperature. The signals were visualized by enhanced chemiluminescence assay using Clarity Western ECL Substrate (Bio-Rad, USA) followed by digital imaging and analysis using the ChemiDoc Imaging System (Bio-Rad, CA). Total protein normalization was employed to assess differences in the levels of the target proteins. The normalized values were obtained by dividing the intensity level of each target protein by the total protein content in the respective sample. This ratio accounts for variations in protein loading among the samples, providing a reliable measure of the target protein expression. The intensity levels were determined using ImageJ (NIH).

### 4.10. Reactive Oxygen Species and Lipid Peroxidation Measurements

For the reactive oxygen species measurements, 0.3 mg of DHE (0.12 mL DMSO and 0.18 mL ddH2O)/30 mg mouse) was injected intraperitoneally (IP), and tissues were harvested after the removal of blood by flushing the left ventricle with PBS (intracardial perfusion) after one hour. Tissues were homogenized with 50 mM phosphate buffer (pH 7.4, 10 μL per mg of tissue) and centrifuged at 12,000× *g* at 4 °C for 8 min. Next, 100 μL of the MeOH/HClO_4_ solution was added to 100 µL of collected supernatant, vortexed for 10 s, and incubated on ice for 1~2 h. The sample was centrifuged at 12,000× *g* at 4 °C for 8 min. Approximately 100 μL of the supernatant obtained was transferred to a fresh tube containing 100 μL of 1 M phosphate buffer (pH 2.6, to induce KClO_4_ salt production). The sample was then vortexed for 5 s. Following this procedure, the sample was centrifuged at 12,000 g at 4 °C for 6 min. Superoxide species in the supernatant were quantified by HPLC. A TBARS/MDA Universal Colorimetric Detection Kit (Arbor Assays, MI; Catalog Number K077-H1) was used to measure malondialdehyde (MDA), a direct product of lipid peroxidation, following the manufacturer’s instructions.

### 4.11. Statistical Analysis

Data are presented as the mean ± S.E.M. Statistical analysis was performed using GraphPad Prism software v9 (GraphPad Software, Inc.). Unless otherwise specified in the figure legend, an ordinary two-way ANOVA with Tukey’s multiple comparisons was used to evaluate the differences between the experimental groups. Statistical differences with *p* < 0.05 were considered significant.

## 5. Conclusions

In conclusion, our data show that restraint stress in mice leads to sexually dimorphic effects on the heart. Previous studies have shown sex differences in gene expression in response to ischemia in the human left ventricular myocardium. In agreement with these findings in humans [51], our results suggest stress-triggered gene expression changes predispose the female heart to a worse outcome after ischemia/reperfusion injury. Additionally, we report for the first time that stress triggers more pronounced changes in ferroptosis gene expression in female hearts, particularly in p53 and p21 levels, which lead to a decrease in Nrf2 and a pro-oxidant environment (elevated ROS and lipid peroxidation). Future studies are warranted to elucidate whether restraint stress, which mimics mental stress in humans, leads to the activation of ferroptosis signaling in the heart. Furthermore, the inclusion of additional markers associated with this specific form of cell death will enhance the validation of the data presented in this study. By identifying the precise mechanism by which stress exacerbates cell death in myocardial infarction in female hearts, valuable insights can be gained regarding potential therapeutic targets that can be translated into human studies.

## Figures and Tables

**Figure 1 ijms-24-10994-f001:**
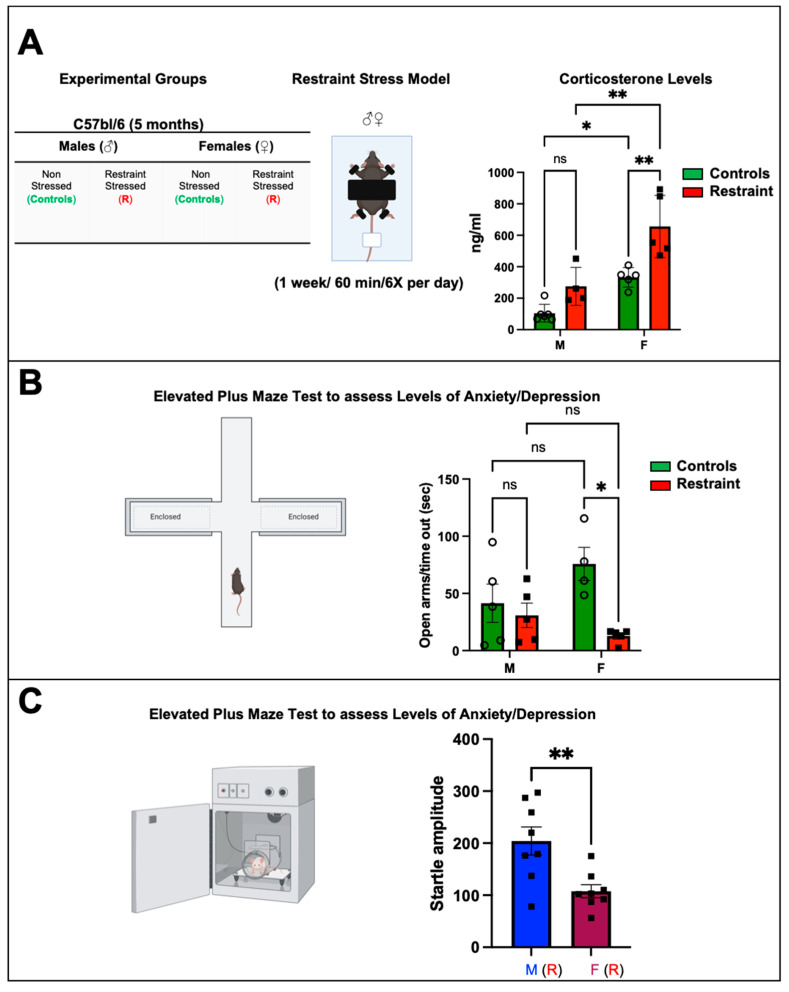
Biochemical and behavioral effects of restraint stress on male and female mice. (**A**) Table showing the experimental groups and schematic representation of the restraint (R) model. Each mouse was placed on the plastic platform, and their upper body and limbs were restrained with double-sided adhesive Velcro tape (Scotch fasteners). The tail was also restrained using nonadhesive medical tape. The head was left unrestrained. Corticosterone was measured. Average corticosterone plasma levels in controls and restraint male (M) and female (F) mice (**B**) Schematic representation of the elevated plus maze. Each mouse was placed on the central platform and allowed to explore the maze for 5 min. Average values for the time in the open arms/time out in seconds after the stress challenge in controls and restraint male and female mice. Individual values for each group are represented as a circle (ο) for males and as a square (■) for females. Controls (green bars) and restraint (red bars). (**C**) Startle response test. Controls and restraint male (M) and female (F) mice were subjected to the prepulse inhibition experimental protocol. Startle amplitudes were detected and measured by a piezoelectric accelerometer mounted directly below the animal’s midline on the Plexiglas frame’s underside. Representation of the average values of the startle amplitude (maximum motion index) for restraint (R) males (blue bars with squares) and restraint (R) females (purple bars with squares). Data represent the mean ± S.E. *n* = 4–8 independent samples per group. * *p* < 0.05, ** *p* < 0.01, ns = not significant. A two-way ANOVA with Tukey’s multiple comparisons analysis was used to evaluate differences among group treatments unless otherwise specified.

**Figure 2 ijms-24-10994-f002:**
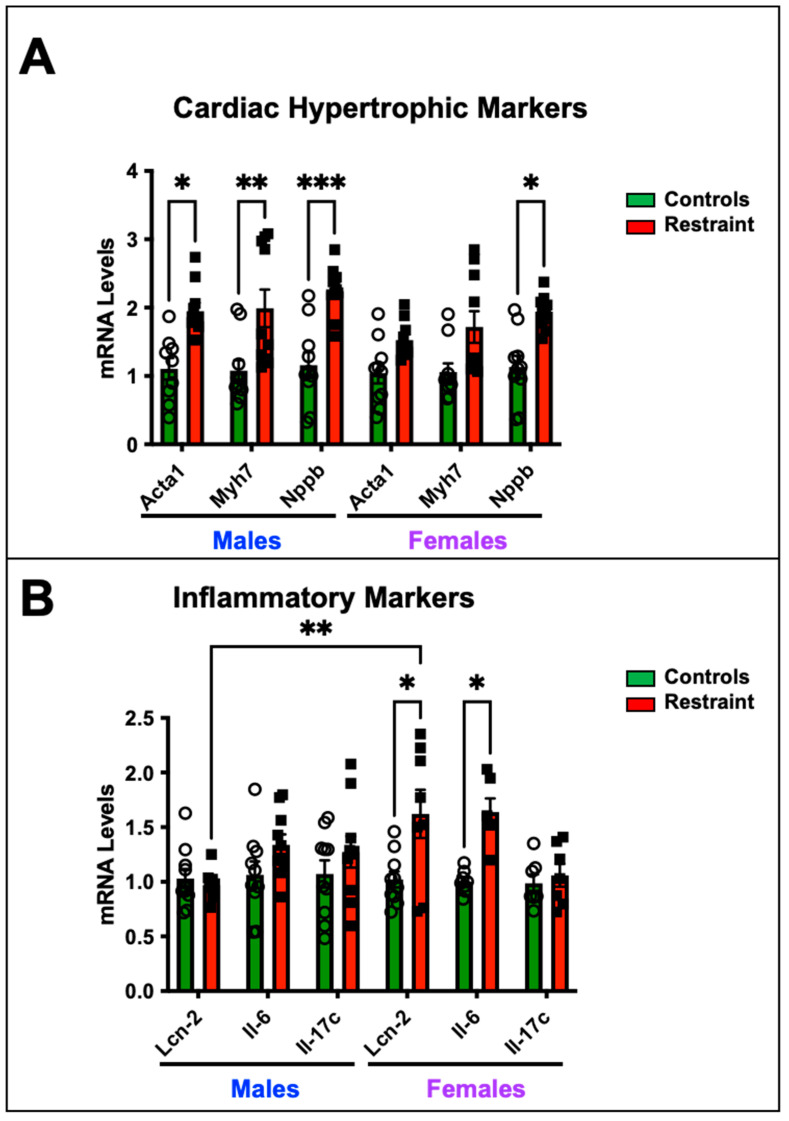
Cardiac expression of hypertrophy and inflammatory markers in response to stress. Total RNA was isolated from whole hearts of 5-month-old nonstressed (controls, green bars) and restraint (red bars) male (M) and female (F) hearts. Individual values for each group are represented as a circle (ο) for males and as a square (■) for females. (**A**) Expression levels (represented as fold-change) of skeletal muscle α-actin (Acta1), β-myosin heavy chain (Myh7), and brain natriuretic peptide (Nppb) were measured by qRT–PCR. (**B**) Expression levels of interleukin-6 (Il-6) and lipocalin 2 (Lcn-2) were measured by qRT–PCR. mRNA was normalized to cyclophilin B (ppib). A two-way ANOVA with Tukey’s multiple comparisons analysis was used to evaluate differences among group treatments unless otherwise specified. Data represent the mean ± S.E. (*n* = 6–10 mice per group). * *p* < 0.05, ** *p* < 0.01, *** *p* < 0.001.

**Figure 3 ijms-24-10994-f003:**
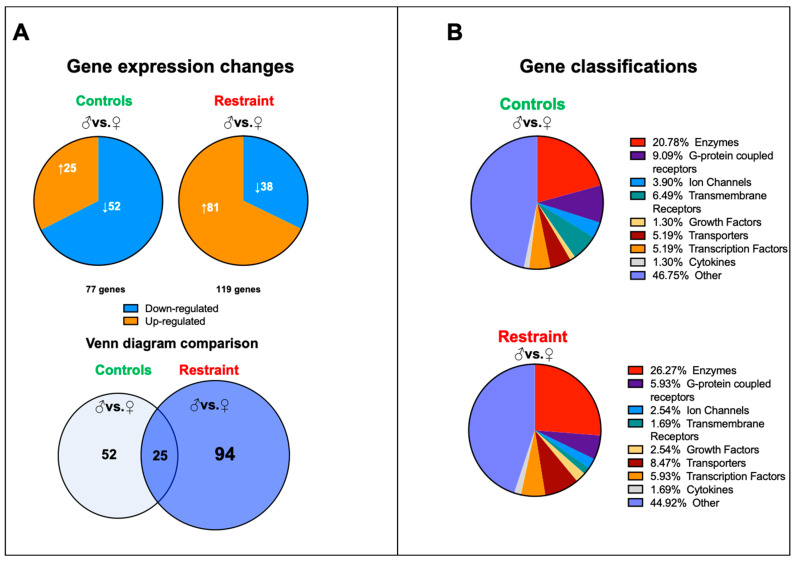
Pattern of gene expression between male and female hearts from controls and restraint mice. (**A**) Analysis of the microarray results showing significantly altered genes in each comparison: control male (♂) to control female (♀) (77 genes) and restraint male to restraint female (119 genes). Orange indicates high-expressing genes, whereas blue indicates low-expressing genes. Venn diagram of the results overlaying controls (males vs. females) and restraint (males vs. females) animals. Only 25 genes are common between control and restraint hearts (intersection). Stress activated the expression of 94 sexually dimorphic genes and abolished the sexually dimorphic expression of 52 genes. (**B**) Gene classifications of the differentially expressed genes. Most sexually dimorphic expressed genes are enzymes and G-protein coupled receptors.

**Figure 4 ijms-24-10994-f004:**
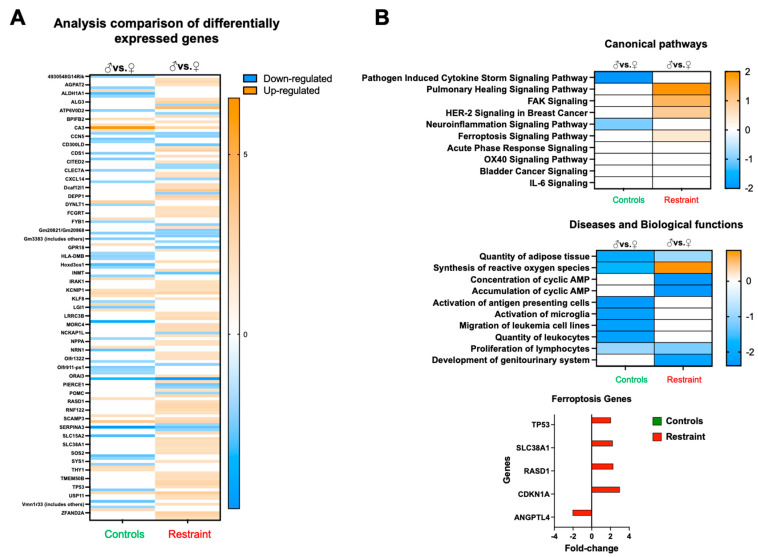
Restraint stress alters gene expression in canonical signaling pathways and other biological functions. (**A**) Heatmap of the microarray data displaying differentially expressed genes between controls and restraint hearts. Orange indicates high-expressing genes, whereas blue indicates low-expressing genes. (**B**) Pathway analysis of the differentially expressed genes. Stress leads to the upregulation of genes involved in a cytokine storm, healing (pulmonary healing signaling pathway), cell migration and growth (FAK signaling and HER-2 [breast cancer] signaling in cancer), neuroinflammation-related genes, and ferroptosis. Genes involved in ferroptosis include tumor protein p53 (p53), solute carrier family 38 member 1 (Slc38A1), dexamethasone-induced Ras related 1 (RasD1), cyclin-dependent kinase inhibitor 1A (Cdkn1A or p21) and angiopoietin-like 4 (Angptl4). Red indicates fold-change in restraint hearts (M vs. F) and green indicates fold-change in control hearts. Pathway and network analyses of the significant genes were conducted using the Ingenuity Pathway Analysis tool (Ingenuity Systems, Redwood City, CA, USA).

**Figure 5 ijms-24-10994-f005:**
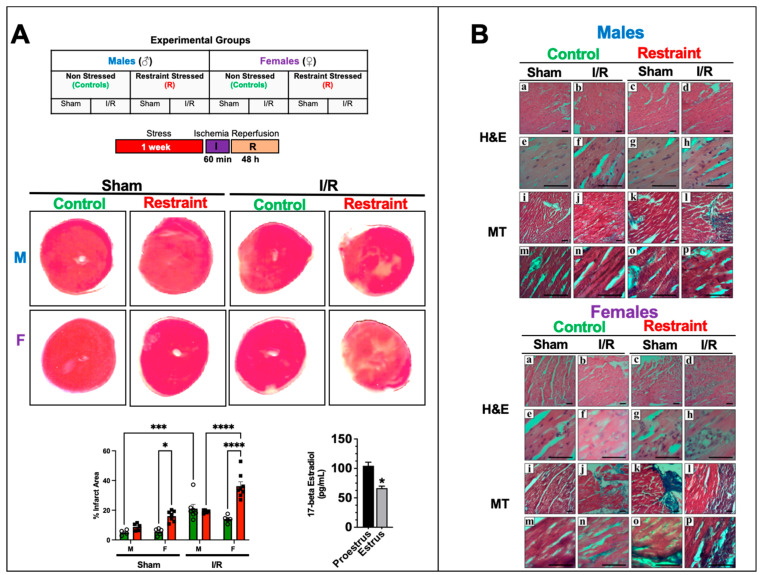
Stress effects on infarct injury after ischemia/reperfusion injury. (**A**) Table showing the experimental groups. Controls and restraint (R) male (M) and female (F) mice were subjected to 60 min of ischemia (I) and 48 h of reperfusion (R) following one week of restraint stress. Representative photographs of triphenyltetrazolium chloride (TTC)-treated heart sections harvested from the sham and ischemia/reperfusion (I/R) groups 48 h after surgery. The infarct areas (%) were quantified using ImageJ color threshold mode. All I/R experiments were performed in females in the proestrus phase of the estrus cycle (highest estrogen activity). Individual values for each group are represented as a circle (ο) for males and as a square (■) for females. Controls (green bars) and restraints (red bars). Plasma levels of 17-beta estradiol in females in proestrus A two-way ANOVA with Tukey’s multiple comparisons analysis was used to evaluate differences among group treatments unless otherwise specified. Data represent the mean ± S.E. *n* = 5–10 mice per group. Only statistically significant differences are shown in the figure. * *p* < 0.05, *** *p* < 0.001, **** *p* < 0.0001. (**B**) Representative micrographs of H&E (upper Panel (**a**–**p**)) and Masson’s trichrome (lower Panel (**a**–**p**)) stained hearts were harvested from controls and restrained males and females. Male and female sections were stained at different times; *n* = 5. The scale bar represents 50 µm.

**Figure 6 ijms-24-10994-f006:**
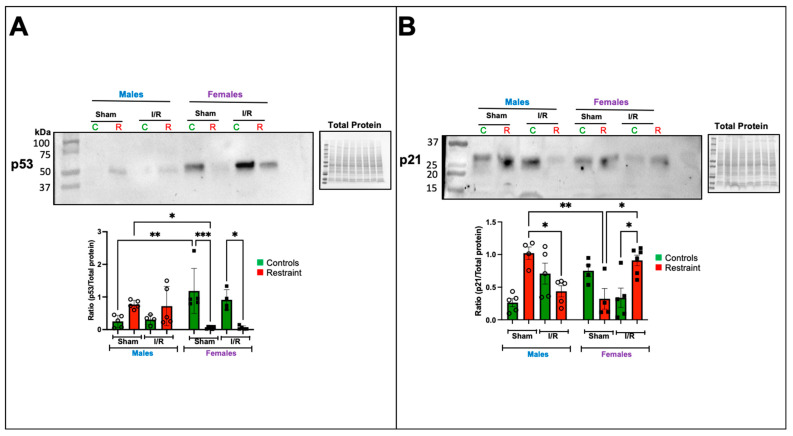
Exposure to stress impacts the transcription and protein expression of ferroptosis-related genes in the heart. Total protein was isolated from whole hearts of 5-month-old control (nonstresssed, green bars) male and female (green bars) hearts and restraint (stressed, red bars) male and female hearts. Protein levels of p53 (**A**) and p21 (**B**) normalized to total protein. A two-way ANOVA with Tukey’s multiple comparisons analysis was used to evaluate differences among group treatments unless otherwise specified. Data represent the mean ± S.E. *n* = 4–6 mice per group. Only statistically significant differences are shown in the figure. * *p* < 0.05, ** *p* < 0.01, *** *p* < 0.001.

**Figure 7 ijms-24-10994-f007:**
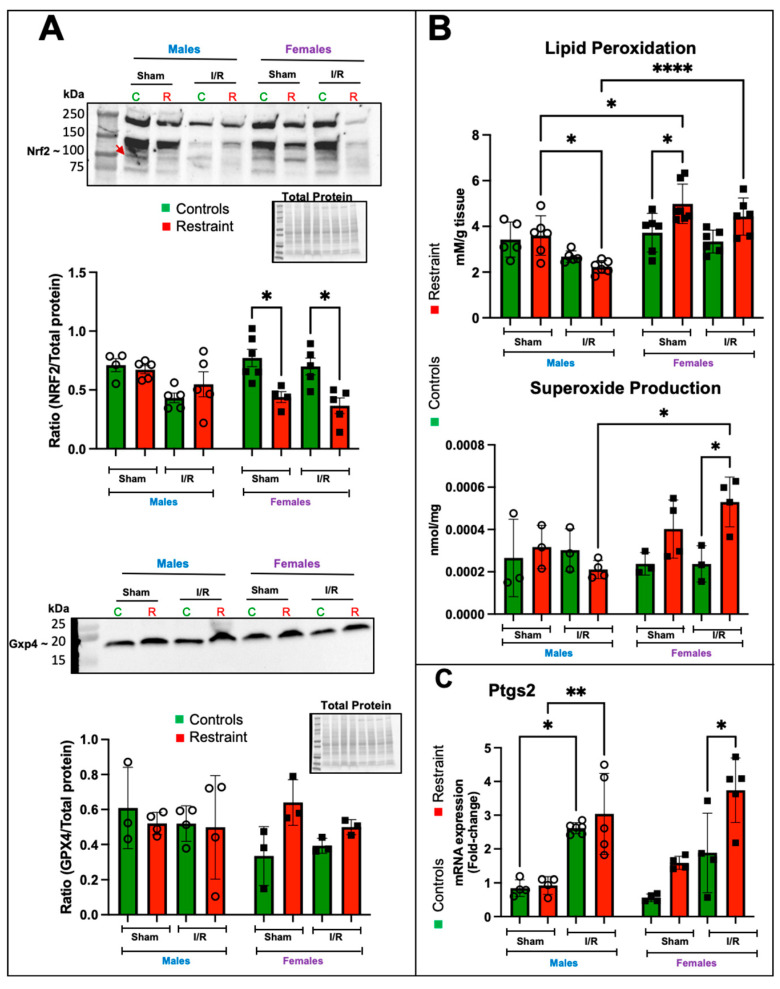
Stress significantly alters the protein levels of Nrf2 and induces increases in lipid peroxidation, superoxide production, and the expression of Ptgs2 in the stressed female heart. Total protein and RNA were isolated from whole hearts of 5-month-old control (green bars) male (M) and female (F) hearts and restraint (red) male and female hearts. Sham-operated animals (without ischemia/reperfusion) and ischemia/reperfusion (I/R). (**A**) Nrf2 (band shown by a red arrow) and Gxp4 protein levels normalized to total protein. (**B**) Lipid peroxidation was measured as malondialdehyde (mM) production per gram (g) of heart tissue (upper figure). Superoxide levels (nmol per mg of heart tissue (middle figure). Prostaglandin-endoperoxide synthase 2 (Ptgs2) mRNA expression levels (**C**). mRNA was normalized to Cyclophillin B (ppib). A two-way ANOVA with Tukey’s multiple comparisons analysis was used to evaluate differences among group treatments unless otherwise specified. Data represent the mean ± S.E. *n* = 3–7 mice per group. Only statistically significant differences are shown in the figure. * *p* < 0.05, ** *p* < 0.01, **** *p* < 0.0001.

## Data Availability

The authors declare that all supporting data and method descriptions are available within the article or from the corresponding authors upon reasonable request.

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
