# Peer review of "Exposure to Stress Alters Cardiac Gene Expression and Exacerbates Myocardial Ischemic Injury in the Female Murine Heart"

_ijms, 2023, doi:10.3390/ijms241310994_

Round 1

Reviewer 1 Report

This study by Dhaibar et al is a great study and has the potential of a landmark paper.  I love it. It shows, that stress has a biochemical different answer in females and males. The authors look at gene expression of some genes, that are altered in ischemia and reperfusion injury. These genes are involved in the regulation of cardiac hypertrophy, inflammation and iron-mediated apoptosis. This is one of the first studies, that is able to indentify factors in the heart, that are differentially regulated in females and males. This study proves: Men's and Women's hearts are different.

However, thare are very few points that need to be adressed:

Fig 1 A: Corticosterone levels in males are more than double in Restraints in females it looks like, that is a little bit lowe than double, but statistically significant, while the difference in males is not significant. Pleas explain in one sentence, that both sexes are able to double their corticosterone levels. Not the differences are diferent, but more the absolute levels of corticosterone. The ACTH levels image is not really necessary, one sentence in the results about this should be ok.

Figure 2: There are no really biolocically relevant data. o.k. the differences in body weight are significant in males nd females, but this is known and obvious. A (very) short description in the text should be enough. Leave this figure out. This is a data-rich story, it does not need this figure.

Figure 3: '(A)Please test the correspondent Restraint markers acta, myh and nppb for significnace in males and females and depict the significance in the image, seem to be higher in males than in females, if not significant, think about leaving these data out or explain it in the text, the teext does not say anything about these markers.

(B) o.k., supported by the text.

Fig 4. no numeric data, but looks colourful and nice, does unfortunately not differentiate between male and female. If possible, please diferntiate controles: male vs femal and Restraint: male vs female. If not possible, leave it as it is. and do not discard this figure.

Fig 5. leave as it is, who is the breast cancer gene upregulated in the rRestraint group? Please explain or leave the HER-2 out.

Fig 6. most important Figure in the paper. Why is in the histology images in the "females" section image b and f in light red, while the others are darker? please make these morre comparable. 

Fig. 7. (A) The "different gene expression diagram is somewhat confusing. Please think about leaving this out. It is not necessary, describe your findings in the text. Enlarge the data images below.

scl38A1 data are not consistant with the story, Restraint Scl38A1 is higher in sham males and in I/R females, the only significance is restraint shams? Try to leave this out. This image does not support the hypothesis. The RASd1 and Angptl1 data are inconsistant as well. Please proved a good explanation in the text or leave out. 

7 (B) p21: The western blot does not reflect the data below, if availabe, show a different blot of another experiment (n=?) or leave the blots out (but show the diagrams). The blot reader might have measured the "blob spots" in this blot. However, the p21 band of male sham control is more intensive than the band of male restraint I/R. In the data it is vice versa. same with femal sham. The bands are not the size of 21 kDa, they are above 25 kDa. Phsphorylation effect? please explain.

8 (A) Western blot: it is not clear, which band was measured. Please spread this blot and clearly mark with an arrow, which band you think is the right one. The two fat bands in thi blot do not reflect the data below. The authors may leave out the blots in '(A') and (B). The diagrams look good and support the hypothesis.

Please adapt the text to the changes in the figures.

Again, congratulations, this is a great paper! 

Author Response

This study by Dhaibar et al is a great study and has the potential of a landmark paper.  I love it. It shows, that stress has a biochemical different answer in females and males. The authors look at gene expression of some genes, that are altered in ischemia and reperfusion injury. These genes are involved in the regulation of cardiac hypertrophy, inflammation and iron-mediated apoptosis. This is one of the first studies, that is able to indentify factors in the heart, that are differentially regulated in females and males. This study proves: Men's and Women's hearts are different.

R= Thank you for your positive comments, enthusiasm, and encouragement to continue pursuing these studies. We have taken your recommendations into consideration and addressed them below. We sincerely appreciate your constructive criticism, as it has greatly improved the quality of our manuscript based on your suggestions.

However, thare are very few points that need to be adressed:

Fig 1 A: Corticosterone levels in males are more than double in Restraints in females it looks like, that is a little bit lowe than double, but statistically significant, while the difference in males is not significant. Pleas explain in one sentence, that both sexes are able to double their corticosterone levels. Not the differences are diferent, but more the absolute levels of corticosterone. The ACTH levels image is not really necessary, one sentence in the results about this should be ok. R=  We agree with the reviewer's observation that corticosterone levels appear to double in both males and females: male controls (104.94 ng/ml ± 56.62) vs. restrained (275.6 ± 54.13 ng/ml), and female controls (332.31 ±27.86 ng/ml) vs. restrained (656.55 ng/ml ± 98.74). However, when we analyzed the data using Two-way ANOVA, we found statistically significant differences only within the female groups. In order to acknowledge the potential biological significance of the increase in corticosterone levels in restrained males, we have included the following statement in the manuscript (Page 7, lines 299-303):

"While restraint stress did not result in a statistically significant increase in corticosterone levels in males, we observed an increase in the average value, indicating that restraint stress in our model leads to biologically significant differences in corticosterone production in both males and females to a similar extent."

We have followed your suggestion and removed the data on ACTH from Figure 1. In the manuscript, we mentioned that ACTH levels were also measured, but no significant differences were found.

Figure 2: There are no really biolocically relevant data. o.k. the differences in body weight are significant in males nd females, but this is known and obvious. A (very) short description in the text should be enough. Leave this figure out. This is a data-rich story, it does not need this figure. R=  We agree with the reviewer's comment, and we have removed Figure 2 and the corresponding section in the Results section.

Figure 3: '(A)Please test the correspondent Restraint markers acta, myh and nppb for significnace in males and females and depict the significance in the image, seem to be higher in males than in females, if not significant, think about leaving these data out or explain it in the text, the teext does not say anything about these markers. R= We have made modifications to Figure 3, which is now Figure 2, and we have denoted statistically significant changes with asterisks (*). Our data reveal that restraint stress induces an increase in the transcription of fetal genes associated with pathological cardiac hypertrophy, such as skeletal muscle α-actin (Ska), β-myosin heavy chain (β-Mhc), and brain natriuretic peptide (Nppb), in the hearts of restrained males compared to their control counterparts (Figure 2A). However, only Nppb exhibited significant up-regulation in the hearts of restrained females compared to control females (Figure 2A). On the other hand, the up-regulation of inflammatory markers, specifically interleukin-6 (Il-6) and lipocalin 2 (Lcn-2), was significantly more pronounced in the hearts of stressed females (S-females) than in their male counterparts (Figure 3B).

(B) o.k., supported by the text.

Fig 4. no numeric data, but looks colourful and nice, does unfortunately not differentiate between male and female. If possible, please diferntiate controles: male vs femal and Restraint: male vs female. If not possible, leave it as it is. and do not discard this figure. R= We have made modifications to Figure 3, and we specify that the comparisons were made between controls (males vs. females) and between restrained individuals (males vs. females).

Fig 5. leave as it is, who is the breast cancer gene upregulated in the rRestraint group? Please explain or leave the HER-2 out. R= HER-2 (breast cancer) refers to genes that are associated with the HER-2 signaling pathway in breast cancer. We have provided clarification regarding this in the figure legend.

Fig 6. most important Figure in the paper. Why is in the histology images in the "females" section image b and f in light red, while the others are darker? please make these morre comparable. R= We agree with this reviewer's comment. In the figure legend, we have clarified that the male and female sections were not stained simultaneously. Regrettably, the sections were stained at different times, which could account for the variation in staining intensity observed. We would like to mention that due to the limited time frame provided by IJSM (only 10 days) for revisions, it was not feasible for us to process new tissue sections. We sincerely hope that the reviewer understands this constraint and considers it while evaluating our revised manuscript.

Fig. 7. (A) The "different gene expression diagram is somewhat confusing. Please think about leaving this out. It is not necessary, describe your findings in the text. Enlarge the data images below.

scl38A1 data are not consistant with the story, Restraint Scl38A1 is higher in sham males and in I/R females, the only significance is restraint shams? Try to leave this out. This image does not support the hypothesis. The RASd1 and Angptl1 data are inconsistant as well. Please proved a good explanation in the text or leave out. R= We agree with this reviewer's comment. These data add confusion. We have removed the data from the revised figure and modified the text to reflect these changes.

7 (B) p21: The western blot does not reflect the data below, if availabe, show a different blot of another experiment (n=?) or leave the blots out (but show the diagrams). The blot reader might have measured the "blob spots" in this blot. However, the p21 band of male sham control is more intensive than the band of male restraint I/R. In the data it is vice versa. same with femal sham. The bands are not the size of 21 kDa, they are above 25 kDa. Phsphorylation effect? please explain. R= The values displayed in the graph represent the average of five blots, which are included in the supplemental materials required by the journal. We have also provided the images below for your review. Regrettably, the image included in the manuscript represents the best representation of the average values depicted in the graph. Due to time constraints in submitting the revised manuscript and the unavailability of additional samples to repeat the experiment, we are unable to conduct the experiment again. Repeating the restraint stress and surgical procedure would require at least one month. We hope the reviewer understands these limitations. Yes, we believe that phosphorylation of p21 may alter its molecular weight. Previous studies have reported a molecular weight range of 23-44 kDa for phospho-p21. We have included this information in the manuscript (Page 17, lines 608-612) to clarify that the detected band size for p21 is higher than the 21 kDa molecular weight.

8 (A) Western blot: it is not clear, which band was measured. Please spread this blot and clearly mark with an arrow, which band you think is the right one. The two fat bands in thi blot do not reflect the data below. The authors may leave out the blots in '(A') and (B). The diagrams look good and support the hypothesis. R= We have added an arrow to indicate the position (molecular weight) at which Nrf2 was detected by the antibody.

Please adapt the text to the changes in the figures. R= All the figures have been revised/modified as suggested.

Again, congratulations, this is a great paper! R= Thank you for your positive feedback and encouragement to continue with this study.

Reviewer 2 Report

The study by Dhaiber et al  show that restraint stress triggers gender prone changes in gene expression related to ferroptosis such that the hearts of female mice showed decreased levels of p53, p21, and Nrf2 downregulation. They claim that this is the first time that stress triggers more pronounced changes in ferroptosis gene expression in female hearts, particularly in p53 and p21 levels. There are some minor comments which might improve the quality of the manuscript.

1.       Please confirm whether it is Visual Sonic 3100 or 31100.

2.       The Ventilation should be mentioned at the beginning of LAD ligation method.

3.       Did the sham animal group undergo reperfusion as mentioned “followed by the same duration of reperfusion as the experimental group?

4.       Previous study indicates that blood gas analysis revealed evidence of a ventilation-perfusion mismatch in the 100% oxygen group. Pressure-volume hysteresis and histomorphometric analyses confirmed the presence of increased atelectasis in mice that received 100% oxygen. How may this finding impact the current results?

5.       The authors described “corticosterone levels in R-females when compared to both control females and R-male subjects. But it was not clear, were the mal control and R group differed. Figure 1A should be described in detail.

6.       In Figures 1A and C how was the statistical test performed. The Figure legend describes this for 1B but not for 1 A and C.

7.       Previous studies have described the role of sex differences in gene expression in MI. The study by Stone et al “PMID: 30649309, report that females have a higher risk of heart failure post-myocardial infarction relative to males. The author should include this finding in their discussion when presenting a conclusion.

8.       How do the results contribute to the survival outcomes in the male and female mice

Author Response

The study by Dhaiber et al  show that restraint stress triggers gender prone changes in gene expression related to ferroptosis such that the hearts of female mice showed decreased levels of p53, p21, and Nrf2 downregulation. They claim that this is the first time that stress triggers more pronounced changes in ferroptosis gene expression in female hearts, particularly in p53 and p21 levels. There are some minor comments which might improve the quality of the manuscript.

  1. Please confirm whether it is Visual Sonic 3100 or 31100. R= Following the recommendation of one of the reviewers, we have excluded the echocardiogram data from the manuscript.
  2. The Ventilation should be mentioned at the beginning of LAD ligation method. R= We have modified the description of the left anterior descending artery ligation procedure as suggested (Page 3, lines 131-135)
  3. Did the sham animal group undergo reperfusion as mentioned “followed by the same duration of reperfusion as the experimental group? R= Yes, the sham animals undergo a 48-hour reperfusion period. We have included this information on page 3, line 153 of the manuscript.
  4. Previous study indicates that blood gas analysis revealed evidence of a ventilation-perfusion mismatch in the 100% oxygen group. Pressure-volume hysteresis and histomorphometric analyses confirmed the presence of increased atelectasis in mice that received 100% oxygen. How may this finding impact the current results? R= Thank you for bringing up these findings. To the best of our knowledge, we did not observe any lung collapse during the surgical procedure. However, we lack data to definitively confirm that this was not a contributing factor in our experiment. Unfortunately, due to the time constraints for resubmitting the manuscript, we are unable to repeat the experiments at this stage to investigate whether mortality during the surgical procedure was indeed caused by atelectasis. We apologize and hope for your understanding.
  5. The authors described “corticosterone levels in R-females when compared to both control females and R-male subjects. But it was not clear, were the mal control and R group differed. Figure 1A should be described in detail. R= We have made modifications to this figure and clarified that only statistically significant differences between the experimental groups are presented. However, it is worth noting that corticosterone levels appear to double in both males and females. In male mice, the control group exhibited levels of 104.94 ng/ml ± 56.62, whereas the restrained group showed levels of 275.6 ± 54.13 ng/ml. Similarly, in female mice, the control group had levels of 332.31 ± 27.86 ng/ml, while the restrained group displayed levels of 656.55 ng/ml ± 98.74. However, when analyzing the data using Two-way ANOVA, we discovered statistically significant differences only within the female groups. To acknowledge the potential biological significance of the increase in corticosterone levels observed in restrained males, we have included the following statement in the manuscript (Page 7, lines 299-303):

"While restraint stress did not result in a statistically significant increase in corticosterone levels in males, we observed an increase in the average value, indicating that restraint stress in our model leads to biologically significant differences in corticosterone production in both males and females to a similar extent."

  1. In Figures 1A and C how was the statistical test performed. The Figure legend describes this for 1B but not for 1 A and C. R= We apologize for the mistake. In Figure 1A-C, we employed a two-way ANOVA with Tukey's multiple comparisons analysis to assess the differences among the treatment groups.
  2. Previous studies have described the role of sex differences in gene expression in MI. The study by Stone et al “PMID: 30649309, report that females have a higher risk of heart failure post-myocardial infarction relative to males. The author should include this finding in their discussion when presenting a conclusion. R= Thank you for the suggestion. The reference has been added (Page 18, line 655).
  3. How do the results contribute to the survival outcomes in the male and female mice. R= We do not possess data regarding the survival outcome of male and female mice following restraint stress and ischemia/reperfusion. It is plausible that exposure to stress could have detrimental effects on survival, particularly if the mice undergo an extended reperfusion period. However, we lack the data necessary to substantiate this hypothesis. We appreciate your understanding.

Reviewer 3 Report

Presented data can be important and useful for the scientific community but there are some issues that have to be resolved.

1) The authors state that the microarray data are available in  GEO database but there is no accession number provided. Please, provide the accession number identifying the data.

2) The experiment involved many analyses but it is not clear whether each animal was involved in all assessments or there were subgroups. Please, specify. If necessary, the authors can provide supplementary table indicating subgroups and applied tests.

3) There is a problem with sample sizes. First of all, the sample sizes differed between groups. What is the reason for these differences? Were some animals / samples rejected? What was the reason for rejecting samples? The 1st figure already indicates discrepancies in number of samples between corticosterone and ACTH assessments.  Furthermore, the information about number of samples seems to be inaccurate. The legend to fig 1 provides information that n = 5-8 but scatter plots indicates smaller number of samples (3-4) in some groups. These problems apply also to other figures.

4) What groups are denoted by S-female and S-male?

5) The authors write: “Differences in BW were only found between control males versus R-males (Figure 2A).” This information is inconsistent with the figure because there are no differences between males.

6) The authors describe differences in expression of some genes in the section” Restraint stress affects the cardiac expression of hypertrophy and inflammatory markers in the absence of significant cardiac function impairments.” Are these results based on microarrays? It is not clear.

7) Western blot images indicate that the authors normalized the results based on an assesment of total proteins but there is no information about this in the methods section.

8) Not all antibodies in methods section are clearly identified.

9) Please, avoid acronyms, they do not help reading the paper.  Please see following paper: https://elifesciences.org/articles/60080

Author Response

Presented data can be important and useful for the scientific community but there are some issues that have to be resolved.

1) The authors state that the microarray data are available in  GEO database but there is no accession number provided. Please, provide the accession number identifying the data. R=

Yes, the data is currently being incorporated into the GEO database. This task is being carried out by the genomics core facility at our institution, and we will obtain the accession number and include it in the manuscript during the proofing stage. We apologize for any inconvenience, but due to the constraints associated with submitting the revisions, we are unable to add it at this time. We appreciate your understanding.

2) The experiment involved many analyses but it is not clear whether each animal was involved in all assessments or there were subgroups. Please, specify. If necessary, the authors can provide supplementary table indicating subgroups and applied tests. R= We have included a table that specifies the experimental groups in the revised Figure 1 and revised Figure 5.

3) There is a problem with sample sizes. First of all, the sample sizes differed between groups. What is the reason for these differences? Were some animals / samples rejected? What was the reason for rejecting samples? The 1st figure already indicates discrepancies in number of samples between corticosterone and ACTH assessments.  Furthermore, the information about number of samples seems to be inaccurate. The legend to fig 1 provides information that n = 5-8 but scatter plots indicates smaller number of samples (3-4) in some groups. These problems apply also to other figures. R= In Figure 1, our sample size ranges from 4 to 8 mice. We sincerely apologize for the mistake and have thoroughly revised all figures to ensure accuracy. Although we did not intentionally reject any samples, it was not feasible to include all animals in every test due to time limitations for equipment usage. Additionally, we encountered technical issues during our experimental procedures, resulting in the loss of some animals. We made every effort to include as many animals as possible in our experiments and aimed to maintain transparency by presenting individual samples in our graphs. Furthermore, based on the suggestion of one of the reviewers, we have removed ACTH from the manuscript.

4) What groups are denoted by S-female and S-male? R= We have removed those acronyms indicating stressed-females and stressed-males.

5) The authors write: “Differences in BW were only found between control males versus R-males (Figure 2A).” This information is inconsistent with the figure because there are no differences between males. R= Following the recommendation of two of the reviewers we have removed the data shown in the original Figure 2 and and the corresponding section in the Results section. 

6) The authors describe differences in expression of some genes in the section” Restraint stress affects the cardiac expression of hypertrophy and inflammatory markers in the absence of significant cardiac function impairments.” Are these results based on microarrays? It is not clear. R= No, these data are not derived from the microarray analysis. To prevent any confusion, we have moved these data under a separate sub-title.

7) Western blot images indicate that the authors normalized the results based on an assesment of total proteins but there is no information about this in the methods section. R= Thank you for bringing this omission to our attention. We have now included the following information in the Methods section (Page 6, lines 266-271):

“Total protein normalization was employed to assess differences in the levels of the target proteins. The normalized values were obtained by dividing the intensity level of each target protein by the total protein content in the respective sample. This ratio accounts for variations in protein loading among the samples, providing a reliable measure of the target protein expression. The intensity levels were determined using ImageJ (NIH).”

8) Not all antibodies in methods section are clearly identified. R= We have included all the information that is available from the manufacturer. All the antibodies used in our study are commercially available.

9) Please, avoid acronyms, they do not help reading the paper.  Please see following paper: https://elifesciences.org/articles/60080 R= We agree with the reviewer's suggestion, and we have made efforts to remove as many acronyms as possible to enhance the readability of the manuscript.

Reviewer 4 Report

In the present study, Dhaibar and colleagues tested the impact of restraint stress alone or combined with myocardial infarction in mice, looking for possible sex-imparted differences. Among the main findings, there is evidence that restraint stress increased corticosterone levels only in females subjected to restraint stress; however, this occurred without a concomitant rise in ACTH. Second, the authors found no substantial differences in cardiac function between genders after restraint stress, except for some gravimetric parameters, such as heart and body weight. The most exciting part of the study comes from the superimposition of restraint stress and myocardial infarction that appears to affect more the female than the male heart. The authors attribute this susceptibility to the possible occurrence of more ferroptosis in the female heart.

I have several major and minor comments to make.

Major Comments

1)      Abstract. What is the central hypothesis of the study? And against what background has it been eventually generated?

2)      Data in Fig.1A are intriguing, but how do the authors explain the surge of corticosterone in females (under control and restrained conditions) without changes in ACTH levels? Are glucocorticoids more elevated in females after stress?

3)      One limitation of the study is that cardiac function was not evaluated on its whole; for instance, it is very plausible that restrained female mice have diastolic dysfunction to begin with; see, for instance, the effects produced by the combination of psychosocial stress and obesity (Agrimi J. et al. EBiomedicine, 2019).

4)      How restraint upregulated the transcription of fetal genes associated with pathological hypertrophy in the heart of females? Same for Il-6 and lipcalin 2.

5)      The second part of the study, i.e., the evaluation of the impact of restrain stress on the functional outcome of myocardial infarction provides the most exciting data of the study. Judging from the data included in Fig.7, it seems that the upregulation of p53 could be at the foundation of increased apoptosis after restraint stress in female MI hearts. However, this hypothesis was not further investigated.

6)      The data reported in Figure 8 are anecdotal in that they do not fully prove that ferroptosis is at play here. This conclusion is inferred from the upregulation of Ptsg2 and the occurrence of lipid peroxidation; however, this evidence should be consolidated by more robust metrics, such as the evaluation of biomarkers such as TFRC and other proteins controlling intracellular iron handling.  

Minor Comments

1)      Introduction. “However, women remain an underrepresented population in cardiovascular (CV) research.”. I would not necessarily subscribe to this contention. My suggestion is to tone down (modify) this sentence.

2)      Introduction. I suggest the Authors limit themselves to enunciating the study's specific questions without anticipating too much the main findings.

3)      The Discussion could be subdivided into paragraphs, each bearing a separate subheading.

Author Response

In the present study, Dhaibar and colleagues tested the impact of restraint stress alone or combined with myocardial infarction in mice, looking for possible sex-imparted differences. Among the main findings, there is evidence that restraint stress increased corticosterone levels only in females subjected to restraint stress; however, this occurred without a concomitant rise in ACTH. Second, the authors found no substantial differences in cardiac function between genders after restraint stress, except for some gravimetric parameters, such as heart and body weight. The most exciting part of the study comes from the superimposition of restraint stress and myocardial infarction that appears to affect more the female than the male heart. The authors attribute this susceptibility to the possible occurrence of more ferroptosis in the female heart. R= Thank you for your positive feedback and constructive criticism. We greatly appreciate the suggested modifications, as they have significantly enhanced the quality of our study.

I have several major and minor comments to make.

Major Comments

1)      Abstract. What is the central hypothesis of the study? And against what background has it been eventually generated? R= We have added our central hypothesis to the abstract as shown below:

“ The central hypothesis of this study is that restraint stress induces sex-specific changes in gene expression in the heart, which leads to an intensified response to ischemia/reperfusion (I/R) injury due to the development of a pro-oxidative environment in female hearts”

2)      Data in Fig.1A are intriguing, but how do the authors explain the surge of corticosterone in females (under control and restrained conditions) without changes in ACTH levels? Are glucocorticoids more elevated in females after stress? R=Based on the suggestion of one of the reviewers, we have removed the data on ACTH from Figure 1. However, in the manuscript, we mentioned that we did measure ACTH levels but did not observe any significant differences. As we explained in the Discussion section (Page 16, lines 510-519), previous studies have demonstrated that elevated glucocorticoid levels activate negative feedback mechanisms mediated by glucocorticoids to terminate the HPA axis response to stress. This feedback inhibits the release of corticotropin-releasing hormone and ACTH from the paraventricular neurons and anterior pituitary gland, respectively. In our model, it is likely that the increased levels of corticosterone signaled back to the pituitary gland, resulting in the repression of ACTH production as part of a negative feedback loop. This feedback mechanism aims to suppress glucocorticoid production by the adrenal gland and regulate the stress response. Therefore, the absence of significant differences in ACTH levels between the control and restraint mice can be attributed to this regulatory process.

3)      One limitation of the study is that cardiac function was not evaluated on its whole; for instance, it is very plausible that restrained female mice have diastolic dysfunction to begin with; see, for instance, the effects produced by the combination of psychosocial stress and obesity (Agrimi J. et al. EBiomedicine, 2019). R= We agree with this reviewer that the evaluation of cardiac function in our study was not comprehensive, and it is possible that diastolic function was significantly compromised in response to restraint stress. Previous studies have provided evidence that mental stress can have detrimental effects on resting left ventricular (LV) diastolic function (Harris KM et al. Impact of Mental Stress and Anger on Indices of Diastolic Function in Patients With Heart Failure. J Card Fail. 2020 Nov;26(11):1006-1010. doi: 10.1016/j.cardfail.2020.07.008; Hieda M et al. Reduced left ventricular diastolic function in women with posttraumatic stress disorder. Am J Physiol Regul Integr Comp Physiol. 2019 Jul 1;317(1):R108-R112. doi: 10.1152/ajpregu.00002.2019).

Regrettably, due to limitations in resources (such as ordering new animals to repeat the experiments) and time constraints for resubmitting the revised manuscript, we are unable to measure diastolic function at this stage in our model. As suggested by another reviewer, we have removed Figure 2, and we have incorporated the following paragraph into the Discussion section (Page 16, lines 521-533) to acknowledge this limitation:

“Another limitation of this study is that the evaluation of cardiac function was not com-prehensive, and it is possible that diastolic function was significantly compromised in response to restraint stress in females. Previous studies have demonstrated that mental stress can adversely affect resting left ventricular (LV) diastolic function in patients with heart failure37 and in women with posttraumatic stress disorder38, therefore, it is plausible that similar changes in diastolic function could be observed in our model. Future studies are necessary to thoroughly investigate the effects of restraint stress on cardiac function and provide a more detailed characterization.”

4)      How restraint upregulated the transcription of fetal genes associated with pathological hypertrophy in the heart of females? Same for Il-6 and lipcalin 2. R= Thank you for bringing this point to our attention. We have included a discussion (Page 16, lines 537-566) below regarding the association between these genes and restraint stress:

"Cardiac hypertrophy serves as an adaptive response to increased workload. However, when this increased workload becomes chronic, the compensatory hypertrophy transitions into pathological hypertrophy. Chronic stress exposure has been linked to an elevated risk of pathological cardiac hypertrophy, primarily due to sustained cardiac workload increments that induce a chronic reactivation of genes associated with fetal development. This reactivation triggers increased protein synthesis, leading to cardiomyocyte growth and remodeling of the cardiac sarcomeres, resulting in structural and mechanical alterations.

β-myosin heavy chain (myh7), skeletal muscle α-actin (Acta1), and brain natriuretic peptide (Nppb) are predominantly expressed in the heart during embryonic and fetal stages, with their expression significantly reduced in ventricles after birth. However, in pathological cardiac hypertrophy, the expression of these genes is markedly upregulated. Previous studies have indicated that imbalances in glucocorticoid signaling in cardiomyocytes are associated with the expression of these markers. Furthermore, their expression shows sexual dimorphism, with higher levels observed in male hearts compared to female hearts. Consistent with these findings, our restraint model exhibited statistically significant increases in myh7, Acta1, and Nppb expression in S-male hearts compared to their non-stressed (NS) counterparts. However, in S-female hearts, only Nppb showed a significant elevation. Published data suggest that GR regulates Lcn2 expression in cardiomyocytes, and alterations in its expression are implicated in the progression to pathological cardiac hypertrophy. In our study, we observed significant upregulation of Il-6 and Lcn2 expression in S-female hearts compared to their NS controls, following restraint stress. These findings indicate that stress may induce cardiac pathology by triggering sex-specific gene expression patterns in the heart."

5)      The second part of the study, i.e., the evaluation of the impact of restrain stress on the functional outcome of myocardial infarction provides the most exciting data of the study. Judging from the data included in Fig.7, it seems that the upregulation of p53 could be at the foundation of increased apoptosis after restraint stress in female MI hearts. However, this hypothesis was not further investigated. R= We agree with the reviewer's suggestion that investigating the role of p53 in our model will provide valuable mechanistic insights into the increased cell death observed in female MI hearts after restraint stress. However, due to time constraints for resubmitting the manuscript, we are unable to conduct additional experiments at this stage. Nevertheless, we have incorporated the following sentences into the Discussion section (Page 18, lines 622-625):

"Restraint stress-induced changes in p53 expression could play a crucial role in mediating the augmented cell death observed in female hearts following ischemia/reperfusion. Further investigations are warranted to elucidate the involvement of p53 signaling in the sex-specific cardiac response to stress."

By including these statements, we highlight the importance of future studies in clarifying the specific role of p53 signaling in the stress-induced response of male and female hearts.

6)      The data reported in Figure 8 are anecdotal in that they do not fully prove that ferroptosis is at play here. This conclusion is inferred from the upregulation of Ptsg2 and the occurrence of lipid peroxidation; however, this evidence should be consolidated by more robust metrics, such as the evaluation of biomarkers such as TFRC and other proteins controlling intracellular iron handling.  R= We acknowledge and agree with the reviewer's comment. While the upregulation of Ptsg2 and the increase in lipid peroxidation and superoxide levels serve as important markers for ferroptosis, they are not the sole indicators of the activation of this particular form of cell death. Conducting additional mechanistic studies is warranted to explore this further. However, due to time and resource constraints, we are unable to measure additional markers of ferroptosis for this manuscript. We hope for your understanding, as we have a limited timeframe for resubmitting our manuscript. We have incorporated the following parragraph at the end of the Discussion section (Page 18, lines 659-666):

“Future studies are warranted to elucidate whether restraint stress, which mimics mental stress in humans, leads to the activation of ferroptosis signaling in the heart. Furthermore, the inclusion of additional markers associated with this specific form of cell death will enhance the validation of the data presented in this study. By identifying the precise mechanism by which stress exacerbates cell death in myocardial infarction in female hearts, valuable insights can be gained regarding potential therapeutic targets that can be translated into human studies.”

Minor Comments

1)      Introduction. “However, women remain an underrepresented population in cardiovascular (CV) research.”. I would not necessarily subscribe to this contention. My suggestion is to tone down (modify) this sentence. R= We have removed this sentence from the introduction section.

2)      Introduction. I suggest the Authors limit themselves to enunciating the study's specific questions without anticipating too much the main findings. R= We have modified the introduction sections as suggested.

 3)      The Discussion could be subdivided into paragraphs, each bearing a separate subheading. R= Thank you for the suggestion; however, we are adhering to the journal's style specifications for this section.

Round 2

Reviewer 3 Report

No other comments.

Reviewer 4 Report

No further comments.